

# Impact of cover crop and mulching on soil physical properties and soil nutrients in a citrus orchard

Tran Van Dung[1], Ngo Phuong Ngoc[2], Le Van Dang[1] and Ngo Ngoc Hung[1]

[1] Soil Science Department, College of Agriculture, Can Tho University, Can Tho, Viet Nam
[2] Department of Plant Physiology-Biochemistry, College of Agriculture, Can Tho University, Can Tho, Viet Nam

## ABSTRACT

**Background:** Cover crops and mulching can ameliorate soil porosity and nutrient availability, but their effects on the physical characteristics and nutrients in the raised bed soils are unclear.

**Methods:** The field experiment was conducted in a pomelo orchard from 2019 to 2021, with an area of 1,500 $m^2$. The treatments included control (no cover crop), non-legume cover crop (*Commelina communis* L.), legume cover crop (*Arachis pintoi* Krabov & W.C. Gregory), and rice straw mulching (*Oryza sativa* L.). At the end of each year (2019, 2020, and 2021), soil samples were collected at four different layers (0–10, 10–20, 20–30, and 30–40 cm) in each treatment. Soil bulk density, soil porosity, and the concentration of nutrients in the soil were investigated.

**Results:** The results revealed that soil bulk density at two depths, 0–10 and 10–20 cm, was reduced by 0.07 and 0.08 $g\ cm^{-3}$ by rice straw mulch and a leguminous cover crop, thus, increasing soil porosity by ~2.74% and ~3.01%, respectively. Soil nutrients (Ca, K, Fe, and Zn) at topsoil (0–10 cm) and subsoil (10–20 cm) layers were not significantly different in the first year, but those nutrients (Ca, K, Fe, and Zn) improved greatly in the second and third years.

**Conclusions:** Legume cover crops and straw mulch enhanced soil porosity and plant nutrient availability (Ca, K, Fe, and Zn). These conservation practices best benefit fruit orchards cultivated in the raised bed soils.

## INTRODUCTION

The loss of nutrients in the soil is considered a key problem for decreasing soil fertility in the fruit orchards grown in raised bed soils (*Dang & Hung, 2021*). In the Vietnamese Mekong Delta (VMD), soil compaction and degradation are more severe (*Dung et al., 2020*). Many studies have reported that reduced soil organic matter is a primary cause of increased soil bulk density (*Athira, Jagadeeswaran & Kumaraperumal, 2019*; *Dang, Ngoc & Hung, 2021*; *Imran, Amanullah & Al-Tawaha, 2022*). Citrus needs high soil porosity and available nutrients for optimum growth and development. Pomelo (*Citrus grandis* Osbeck)

Corresponding author
Ngo Ngoc Hung,
ngochung@ctu.edu.vn

has been cultivated in many places at the VMD, and they are a great source of income for growers (*Kieu & Hung, 2019*). However, the pomelo productivity cultivated on old raised soils has been reduced due to poor soil fertility and compaction (*Dung et al., 2020*). *Dang, Ngoc & Hung (2022)* reported that soil acidity in the citrus orchards increased significantly with chemical fertilizers (ammonium sulfate, superphosphate, and potassium chloride) in the long term. Furthermore, farmers in these areas were not interested in implementing soil conservation measures in their orchards. This may decrease soil moisture, water use efficiency, and biological activity (*Cao et al., 2021*; *Paul et al., 2021*).

Soil conservation practices (mulching, cover cropping, crop rotation, *etc.*) are measures the farmer can apply to mitigate soil degradation and soil erosion (*Ogunsola, Adeniyi & Adedokun, 2020*; *López-Vicente et al., 2020*). Soil conservation measures reduce soil loss by keeping a cover over the ground, decreasing soil displacement associated with raindrops and irrigation water affecting soil particles (*Vincent-Caboud et al., 2019*; *Calegari et al., 2020*). Additionally, soil conservation measures decrease the pressure and velocity of runoff on the topsoil (*Kumawat et al., 2020*). According to *Page, Dang & Dalal (2020)*, conservation practices improved the soil's organic carbon content, soil porosity, water capacity, plant nutrient availability, soil biota activity, and crop productivity.

Cover cropping is a cropping system/method utilized to decrease erosion, ameliorate soil porosity, enhance soil organic matter, weed control, pests, and disease management, and increase biodiversity (*Sharma et al., 2018*; *Das, Kandpal & Devi, 2021*). According to *Van Sambeek (2017)* and *Abdalla et al. (2019)*, cover crops attract pollinators leading to improved fruit set ratio, thus increasing plant productivity. There are two key cover crops: legumes and non-legumes (*Abdalla et al., 2019*). Leguminous cover crops increase soil nutrients especially total and available nitrogen, because they can fix nitrogen biologically (*Kaye et al., 2019*; *MacMillan et al., 2022*; *Freidenreich et al., 2022*). Meanwhile, the nonlegume cover crops increase crop biomass and decrease soil loss from the surface layer (*Romdhane et al., 2019*; *Amanullah et al., 2021*).

Mulches comprising organic materials (straw, litter, *etc.*) covered the soil surface to control weeds and reduce runoff (*Li, Li & Pan, 2021*; *Khoramizadeh et al., 2021*). Mulches help increase soil organic carbon, decreasing soil compaction (*Iqbal et al., 2020*). The decomposition process of organic mulches releases many nutrients (*Khoramizadeh et al., 2021*). These nutrients are in a form that is useful to plants and might increase the uptake, improving crop productivity (*Singh et al., 2021*). Mulching also affects soil microorganism activity and abundance (*Rodrigues da Silva et al., 2022*).

A previous study indicated that cover cropping with legumes and rice straw mulch significantly increases soil organic carbon, total nitrogen, and phosphorus (*Dung et al., 2022*). However, the effects of conservation practices on soil compaction and available nutrients (Ca, Mg, K, Cu, Fe, Zn, and Mn) have not been reported. Hence, this study aimed to evaluate soil conservation measures on soil bulk density, porosity, and soil nutrients in a pomelo orchard cultivated on alluvial soil of the Mekong Delta, Vietnam.

**Table 1 Basic soil physicochemical properties at the study location.**

| Depth (cm) | pH$_{H2O}$ | SOM (%) | Macronutrients (cmol$_c$ kg$^{-1}$) | | | Trace elements (mg kg$^{-1}$) | | | | BD (g cm$^{-3}$) |
|---|---|---|---|---|---|---|---|---|---|---|
| | | | Ca$^{2+}$ | K$^+$ | Mg$^{2+}$ | Cu | Fe | Zn | Mn | |
| 0–10 | 5.02 | 1.50 | 3.53 | 0.16 | 2.28 | 22.7 | 8.25 | 55.1 | 28.6 | 1.19 |
| 10–20 | 4.95 | 1.42 | 3.29 | 0.18 | 2.36 | 30.5 | 8.36 | 45.2 | 24.2 | 1.22 |
| 20–30 | 5.25 | 1.35 | 4.10 | 0.21 | 2.32 | 26.9 | 7.45 | 39.5 | 30.1 | 1.25 |
| 30–40 | 5.18 | 1.20 | 3.98 | 0.17 | 2.41 | 27.0 | 6.32 | 40.3 | 25.7 | 1.23 |

## MATERIALS AND METHODS

### Study site, soil, and climate

A pomelo orchard used for the experiment in this research was the same as described in our previous study (*Dung et al., 2022*). It was located in Hau Giang province (9°54′30.3″N, 105°51′06.7″E). The soil was classified as Gleyic Anthrosols based on the reference of *World Reference Base for Soil Resources (2015)*.

The average monthly rainfall from 2019 to 2021 at the study site was 175 mm, with September and March usually receiving the highest (470 mm) and lowest (10 mm) rainfalls, respectively. Table 1 shows the physical and chemical properties.

### Experimental design

The field experiment was arranged in a randomized complete block design, including four treatments. Each treatment had four replications. The treatments were no cover crop (control), nonlegume (*Commelina communis* L.) cover crop (NLC), legume (*Arachis pintoi* Krabov & W.C. Gregory) cover crop (LCC), and rice straw (*Oryza sativa* L.) mulching (RSM). The number of trees per trial plot was three plants. The five-year-old "Da Xanh" pomelo orchard was used in this study, with an average fruit yield of 18 t ha$^{-1}$ year$^{-1}$. The pomelo plants were 3.0–3.4-m tall at the beginning study, and the canopy diameter was 2.8–3.1 m. All treatments accepted the no-till practice. All treatments applied the same amount of NPK fertilizers at rates of 400 kg N, 300 kg P$_2$O$_5$, and 400 kg K$_2$O ha$^{-1}$ year$^{-1}$ (*Dang, Ngoc & Hung, 2022*).

Nicotex Co., Ltd. (Thai Binh, Vietnam), a commercial product, was used for weed management in the control plots. The herbicide with commercial name NIPHOSATE 480SL contains 480-g glyphosate IPA salt per liter. The spraying rate was 2.5 liter per ha, per the producer's recommendation. A hand sprayer (Mitsuyama TL–767) was used for herbicide application. The weeds were regularly controlled when they reached ~8–10-cm tall (~5–6 leaves), and ~3 months of herbicide was applied.

Asiatic dayflower (*Commelina communis* Krabov & W.C. Gregory) was utilized for NLC plots. Asiatic dayflower was cultivated by cuttings that were ~20-cm long. When the Asiatic dayflower was >30-cm high, they cut the tops by ~20 cm using the Honda Grass Cutter GX35. Pinto peanut (*Arachis pintoi*) was used for LCC plots. The pinto peanut was cultivated by clusters of 2–3 cuttings spaced 10–15-cm apart.

The method and timing of rice straw mulch were in accordance with the Southern Horticultural Research Institute (SOFRI) recommendation, Vietnam. The VMD is located in a tropical monsoon area, and the annual highest rainfall was from June to September. Hence, rice straw mulch was regularly carried out twice yearly in the dry season (October and March). When rice straw mulch was conducted in the wet season, it decreased soil drainage capacity, resulting in increased root rot disease of pomelo (*Thu et al., 2018*).

Under the pomelo tree canopy, the ridges were mulched with a rice straw that was 1.5-m wide and 2–2.5-cm thick. Rice straw used for the experiment was 5.5 t ha$^{-1}$ year$^{-1}$, and an average of ~8.8 kg of rice straw tree$^{-1}$ year$^{-1}$.

## Soil collection and analysis

### Soil physical parameters

In order to determine soil bulk density (BD), soil sample rings Eijkelkamp company were used for soil sampling in 2019, 2020, and 2021. The soil sample ring was 51 mm in height and 53 mm in diameter. Five soil samples were randomly taken from each plot for the BD analysis. After collection, soil cores were dried at 100 °C for 48 h in an oven. BD was calculated from the dry soil mass ratio per unit volume of the soil cores (*Mtyobile, Muzangwa & Mnkeni, 2020*). The total porosity of the soil was calculated from the soil BD values and the particle density. In this study, particle density is 2.65 g cm$^{-1}$. The total porosity is shown in the following equation:

$$\text{Total porosity (\%)} = 1 - \frac{(\text{Soil bulk density})}{2.65} \times 100 \qquad (1)$$

### Soil chemical parameters

In each plot, a soil auger took five soil cores from depths of 0–10 cm, 10–20 cm, 20–30 cm, and 30–40 cm, following a zig-zag pattern in 2019, 2020, and 2021. The five samples from the same depth were blended into one composite sample per depth. The soil was then divided into subsamples of ~500 g. All soil samples were air-dried and ground to pass through a 2 mm sieve.

A 0.1 M BaCl$_2$ extraction was used to analyze the exchangeable base cations (K, Ca, and Mg) (*Hendershot & Duquette, 1986*). The soil's iron content was extracted in oxalate–oxalic acid (*Novozamsky et al., 1986*). Nitric–perchloric acid digestion was performed on Mn, Cu, and Zn, following the procedure recommended by the *AOAC (1990)*. The macroelements (K, Ca, and Mg) and micronutrients (Fe, Mn, Cu, and Zn) were determined using Atomic Absorption Spectrometers (Thermo Scientific$^{TM}$ iCE$^{TM}$ 3000 Series; Thermo Scientific, Waltham, MA, USA).

## Statistics

The statistical analysis relied on SPSS version 20.0. Analysis of variance was used to compare the differences between means among treatments by the Duncan test at a statistical level of $P < 0.05$ (*) and $P < 0.01$ (**).

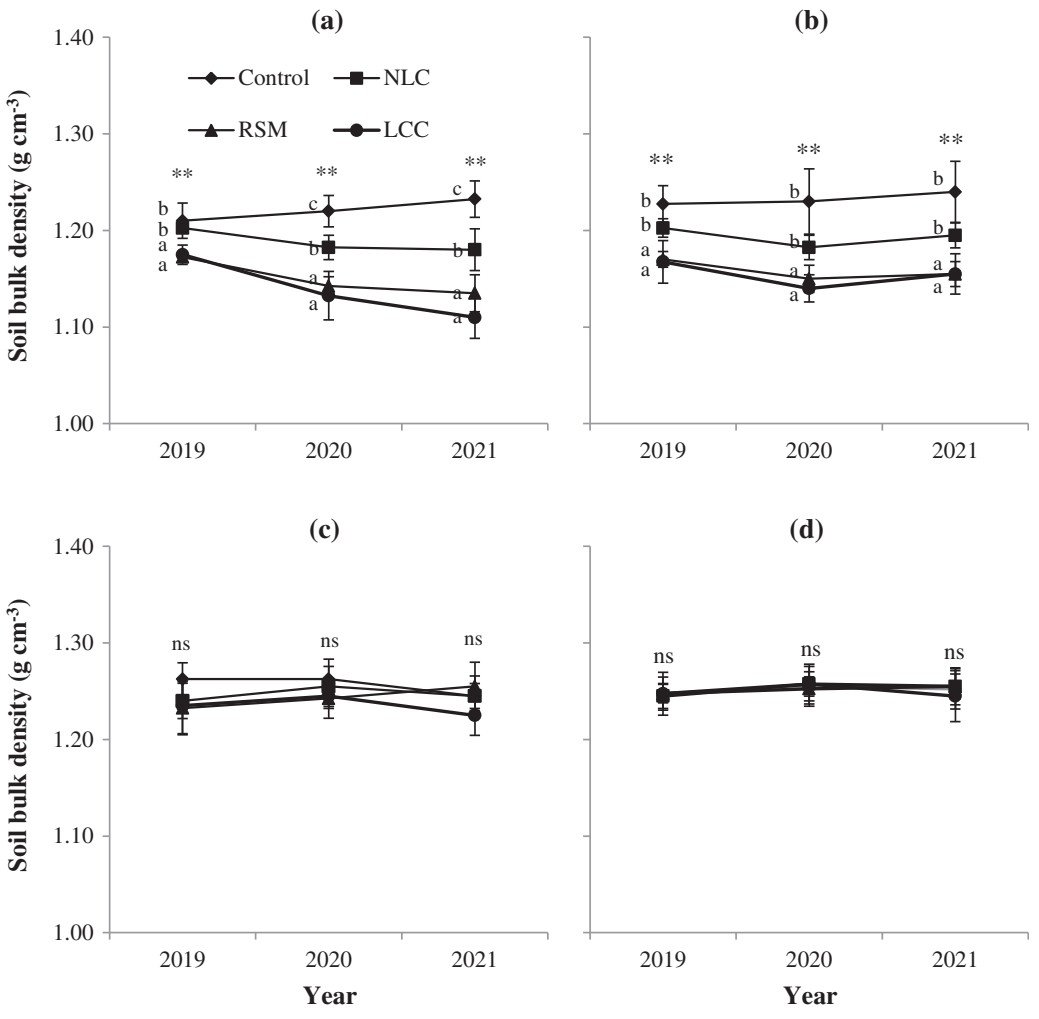

**Figure 1 Soil bulk density is influenced by soil conservation practices: (A) 0–10 cm, (B) 10–20 cm, (C) 20–30 cm, (D) 30–40 cm.** Different letters show a significant difference at $P < 0.01$ (**); ns is not significant. Error bars represent the standard deviation ($n = 4$). Control, no conservation practices; NLC, non-legume cover crop; RSM, rice straw mulching; LCC, legume cover crop.

# RESULTS

## Effect of soil conservation practices on soil bulk density

Figure 1 shows that using soil conservation practices (LCC and RSM) significantly improved BD at both 0–10- and 10–20-cm after 3 years of experimentation. However, soil conservation measures did not affect BD at two depths (20–30 and 30–40 cm). At the topsoil (0–10 cm), BD in LCC and RSM treatments were lower than in the control and NLC plots. In particular, the mean value of BD in LCC and RSM was ~1.14 and ~1.15 g cm$^{-3}$, respectively, lower than that in the treatments of control 1.22 g cm$^{-3}$.

Using of NLC positively affected BD in the topsoil (0–10 cm) in 2020 and 2021 compared with the control treatment (Fig. 1A). Similarly, BD was reduced by covering crops with pinto peanuts and mulching with rice straw in the 10–20 cm soil depth

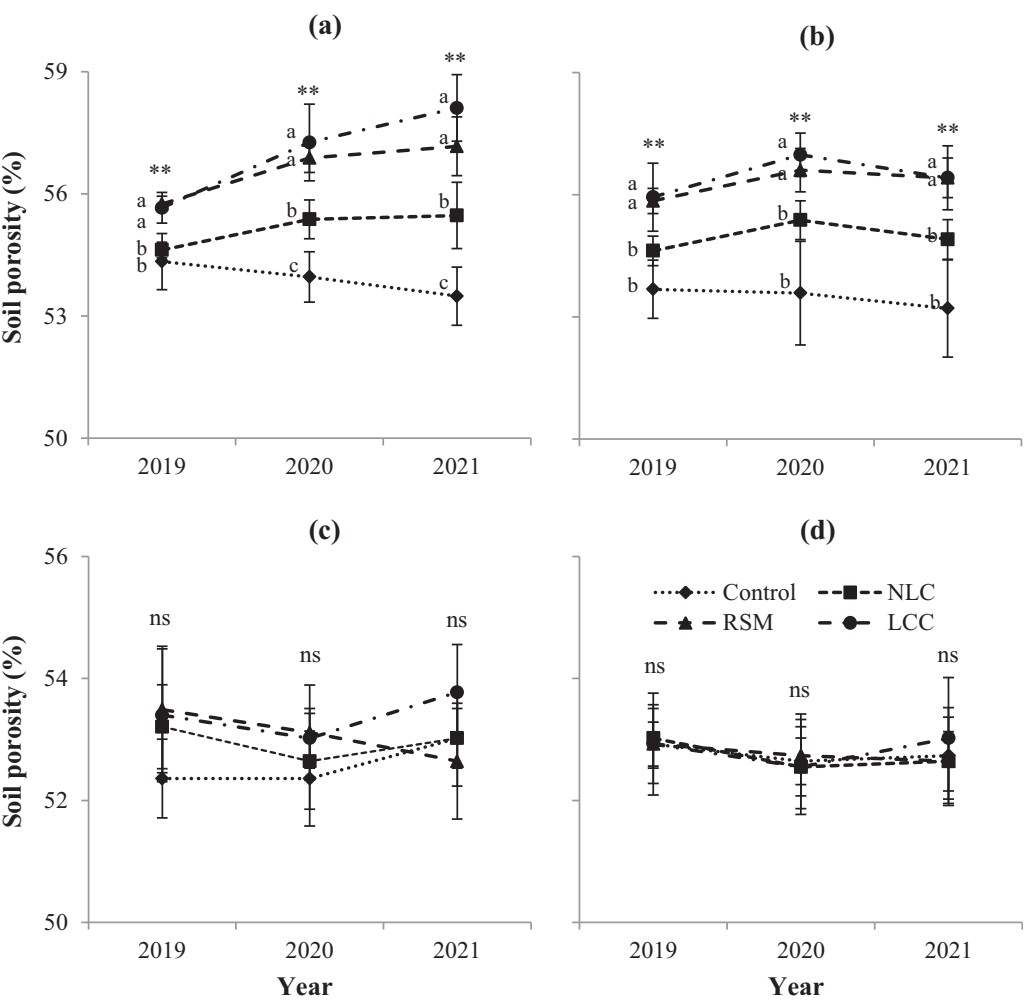

**Figure 2 Soil porosity is affected by soil conservation practices: (A) 0–10 cm, (B) 10–20 cm, (C) 20–30 cm, (D) 30–40 cm.** Different letters show a significant difference at $P < 0.01$ (\*\*); ns is not significant. Error bars represent the standard deviation ($n = 4$). Control, no conservation practices; NLC, non-legume cover crop; RSM, rice straw mulching; LCC, legume cover crop.

(Fig. 1B). Meanwhile, Figs. 1A and 1B showed that BD in the lower layers was not changed after soil conservation measures application. The value of BD in two depths (20–30 cm and 30–40 cm) ranged from 1.23–1.26 g cm$^{-3}$.

## Soil porosity is affected by soil conservation measures

Soil conservation measures increased soil porosity at two depths, 0–10 cm and 10–20 cm (Fig. 2). Like BD, non-legume or legume cover crops and RSM did not improve soil porosity in deeper soil layers (20–30 cm and 30–40 cm). The use of conservation practices (LCC and RSM) enhanced soil porosity by ~5% and ~3% at 0–10 and 10–20 cm (Figs. 2A, 2B) after 3 years of experiments, respectively. In the depths of 20–30 and 30–40 cm, there was no significant difference in soil porosity between soil conservation measures compared to no conservation (Figs. 2C, 2D).

**Table 2 Effect of soil conservation practices on nutrients availability in topsoil layer (0–10 cm).**

| Years | Treatments | Macronutrients (cmol$_c$ kg$^{-1}$) | | | Trace elements (mg kg$^{-1}$) | | | |
|---|---|---|---|---|---|---|---|---|
| | | Ca$^{2+}$ | K$^+$ | Mg$^{2+}$ | Cu | Fe | Zn | Mn |
| 2019 | Control | 3.55 | 0.16 | 2.28 | 25.8 | 8.37 | 59.8 | 26.7 |
| | NLC | 3.52 | 0.17 | 2.30 | 26.7 | 8.63 | 58.0 | 26.9 |
| | RSM | 3.51 | 0.18 | 2.27 | 25.7 | 9.07 | 59.0 | 27.1 |
| | LCC | 3.54 | 0.17 | 2.26 | 24.9 | 8.70 | 59.5 | 27.1 |
| | $P$-value | ns | ns | ns | ns | ns | ns | ns |
| 2020 | Control | 3.45b | 0.15c | 2.27 | 25.2 | 8.57b | 53.1c | 27.6 |
| | NLC | 3.60b | 0.19b | 2.34 | 26.5 | 10.2b | 59.6b | 27.8 |
| | RSM | 3.76a | 0.23ab | 2.30 | 26.3 | 13.6a | 64.8ab | 26.7 |
| | LCC | 3.74a | 0.24a | 2.30 | 27.5 | 13.4a | 66.5a | 26.8 |
| | $P$-value | * | ** | ns | ns | ** | ** | ns |
| 2021 | Control | 3.47c | 0.14c | 2.33 | 26.2 | 8.79c | 58.0b | 27.1 |
| | NLC | 3.71b | 0.23b | 2.36 | 24.5 | 12.2b | 65.7b | 26.0 |
| | RSM | 3.86a | 0.27a | 2.29 | 24.8 | 15.4a | 72.4a | 26.0 |
| | LCC | 3.85a | 0.28a | 2.37 | 26.1 | 16.5a | 72.9a | 26.3 |
| | $P$-value | ** | ** | ns | ns | ** | ** | ns |

Note:
Control, no conservation practices; NLC, non-legume cover crop; RSM, rice straw mulching; LCC, legume cover crop. Different letters in each column indicate significant differences among treatments at $P < 0.05$ (*) and $P < 0.01$ (**); ns, not significant.

## Influence of soil conservation practices on soil nutrients

### Topsoil layer (0–10 cm)

Although the concentrations of macroelements (Ca, K, and Mg) in soil did not improve in the first year of applying conservation practices, they increased significantly in the next two years, except for Mg (Table 2). In particular, the Ca content in the RSM treatments increased by 0.31 and 0.39 cmol$_c$ kg$^{-1}$ in 2020 and 2021 compared with the control, respectively. Those in the LCC treatment were 0.29 and 0.38 cmol$_c$ kg$^{-1}$. Likewise, the K concentration in RSM and LCC was enhanced by ~0.11 and ~0.12 cmol$_c$ kg$^{-1}$ in 3 years of experimentation. Conversely, using the cover crop or mulching did not affect the concentration of Mg in soil. The application of soil conservation measures did not affect the micronutrients (Cu, Fe, Zn, and Mn) contents in 2019 (Table 2). However, the concentrations of Fe and Zn in 2021 were elevated by ~7.0 and ~13.0 mg kg$^{-1}$ compared to 2019 because the crops were covered with legumes and mulched with rice straw. The difference in Fe and Zn improvement might be from Fe and Zn contents containing the legume and rice straw, resulting in enhanced soil Fe and Zn concentration. Soil conservation practices did not influence the contents of Cu and Mn.

### Subsurface layer (10–20 cm)

Table 3 indicates the effect of cover crops and organic mulching on soil fertility. In 2019, soil nutrients (Ca, K, Mg, Cu, Fe, Zn, and Mn) were not increased by soil conservation practices, except for Zn. LCC significantly increased exchangeable Ca by 0.61 and 0.72 cmol$_c$ kg$^{-1}$ compared with control in 2020 and 2021, respectively. Exchangeable Ca

**Table 3 The availability of plant nutrients influenced by conservation agriculture in subsurface layer (10–20 cm).**

| Years | Treatments | Macronutrients (cmol$_c$ kg$^{-1}$) | | | Trace elements (mg kg$^{-1}$) | | | |
|---|---|---|---|---|---|---|---|---|
| | | Ca$^{2+}$ | K$^+$ | Mg$^{2+}$ | Cu | Fe | Zn | Mn |
| 2019 | Control | 3.43 | 0.18 | 2.41 | 27.5 | 9.66 | 49.4b | 27.0 |
| | NLC | 3.51 | 0.19 | 2.48 | 26.0 | 9.76 | 61.8a | 25.6 |
| | RSM | 3.50 | 0.19 | 2.50 | 25.6 | 9.72 | 62.0a | 25.6 |
| | LCC | 3.51 | 0.19 | 2.54 | 27.1 | 9.72 | 64.0a | 26.7 |
| | $P$-value | ns | ns | ns | ns | ns | ** | ns |
| 2020 | Control | 3.42c | 0.17b | 2.35 | 27.0 | 8.98c | 52.6b | 26.6 |
| | NLC | 3.72b | 0.22ab | 2.37 | 26.1 | 11.6b | 62.4a | 25.6 |
| | RSM | 3.91ab | 0.24a | 2.37 | 27.5 | 13.4a | 65.7a | 26.0 |
| | LCC | 4.03a | 0.25a | 2.32 | 26.6 | 14.0a | 65.5a | 26.3 |
| | $P$-value | ** | * | ns | ns | ** | * | ns |
| 2021 | Control | 3.41b | 0.18b | 2.41 | 27.2 | 9.11b | 55.5b | 26.5 |
| | NLC | 3.93a | 0.24a | 2.37 | 26.2 | 13.3a | 62.3a | 25.0 |
| | RSM | 4.10a | 0.28a | 2.33 | 25.9 | 14.1a | 65.4a | 26.8 |
| | LCC | 4.13a | 0.27a | 2.41 | 26.2 | 15.1a | 65.7a | 26.1 |
| | $P$-value | ** | ** | ns | ns | ** | * | ns |

**Note:**
Control, no conservation practices; NLC, non-legume cover crop; RSM, rice straw mulching; LCC, legume cover crop. Different letters in each column indicate significant differences among treatments at $P < 0.05$ (*) and $P < 0.01$ (**); ns, not significant.

was significantly higher in RSM than in control. The exchangeable K$^+$ was higher by an average of 0.07–0.10 cmol$_c$ kg$^{-1}$ in RSM and LCC than in control in 2020 and 2021. Available Fe concentrations were ~1.5-fold greater in LCC and RSM than in no conservation treatment in 2 years (Table 3). Similarly, RSM and LCC enhanced available Zn by more than 10 mg kg$^{-1}$ compared with control in the experiment of 3 years. In the current research, soil conservation practices did not affect the concentrations of Mg, Cu, and Mn.

### The layer of 20–30 cm

In a 3-year study, soil conservation practices did not improve soil quality at a depth of 20–30 cm (Table 4). However, in 2021, the concentration of Cu was the highest in LCC, followed by NLC, RSM, and control. The value of macronutrients (Ca, K, Mg) ranged from 4.00–4.22 cmol$_c$ kg$^{-1}$, 0.18–0.22 cmol$_c$ kg$^{-1}$, and 2.31–2.47 cmol$_c$ kg$^{-1}$, respectively. There was no significant difference in all treatments for micronutrient (Fe, Zn, and Mn) concentrations for micronutrient (Fe, Zn, and Mn) concentrations. Fe, Zn, and Mn concentrations were 8.71–11.3 mg kg$^{-1}$, 38.8–45.9 mg kg$^{-1}$, and 24.3–30.4 mg kg$^{-1}$ from 2019 to 2021.

### The layer of 30–40 cm

The results in Table 5 showed no significant differences in all treatments regarding soil chemical properties, except exchangeable K in 2021 was influenced by soil conservation

**Table 4 Influence of soil conservation practices on macro-micronutrients in the soil at a depth of 20–30 cm.**

| Years | Treatments | Macronutrients (cmol$_c$ kg$^{-1}$) | | | Trace elements (mg kg$^{-1}$) | | | |
|---|---|---|---|---|---|---|---|---|
| | | Ca$^{2+}$ | K$^+$ | Mg$^{2+}$ | Cu | Fe | Zn | Mn |
| 2019 | Control | 4.15 | 0.19 | 2.31 | 24.4 | 8.71 | 39.5 | 26.2 |
| | NLC | 4.15 | 0.18 | 2.41 | 26.9 | 8.94 | 39.5 | 25.6 |
| | RSM | 4.09 | 0.19 | 2.36 | 23.9 | 8.79 | 43.4 | 27.2 |
| | LCC | 4.10 | 0.21 | 2.36 | 23.8 | 8.93 | 44.3 | 25.4 |
| | $P$-value | ns | ns | ns | ns | ns | ns | ns |
| 2020 | Control | 4.00 | 0.20 | 2.38 | 27.4 | 9.67 | 40.5 | 30.4 |
| | NLC | 4.22 | 0.18 | 2.46 | 25.8 | 10.0 | 39.5 | 28.5 |
| | RSM | 4.17 | 0.21 | 2.45 | 24.1 | 9.93 | 38.8 | 28.3 |
| | LCC | 4.06 | 0.22 | 2.47 | 23.7 | 10.7 | 43.2 | 29.2 |
| | $P$-value | ns | ns | ns | ns | ns | ns | ns |
| 2021 | Control | 4.05 | 0.19 | 2.33 | 24.2b | 10.3 | 44.7 | 26.2 |
| | NLC | 4.11 | 0.19 | 2.45 | 24.3b | 10.9 | 42.0 | 25.8 |
| | RSM | 4.07 | 0.19 | 2.31 | 23.9b | 11.3 | 45.9 | 25.5 |
| | LCC | 4.03 | 0.18 | 2.42 | 27.8a | 10.0 | 41.8 | 24.3 |
| | $P$-value | ns | ns | ns | * | ns | ns | ns |

Note:
Control, no conservation practices; NLC, non-legume cover crop; RSM, rice straw mulching; LCC, legume cover crop. Different letters in each column indicate significant differences among treatments at $P < 0.05$ (*); ns, not significant.

**Table 5 Effect of soil conservation measures on availability of plant nutrients at a depth of 30–40 cm.**

| Years | Treatments | Macronutrients (cmol$_c$ kg$^{-1}$) | | | Trace elements (mg kg$^{-1}$) | | | |
|---|---|---|---|---|---|---|---|---|
| | | Ca$^{2+}$ | K$^+$ | Mg$^{2+}$ | Cu | Fe | Zn | Mn |
| 2019 | Control | 3.98 | 0.17 | 2.33 | 25.3 | 5.72 | 48.9 | 25.7 |
| | NLC | 4.02 | 0.17 | 2.33 | 24.2 | 5.79 | 47.0 | 25.7 |
| | RSM | 3.88 | 0.18 | 2.39 | 25.5 | 5.94 | 49.2 | 25.4 |
| | LCC | 4.09 | 0.18 | 2.34 | 23.7 | 5.61 | 49.5 | 24.7 |
| | $P$-value | ns | ns | ns | ns | ns | ns | ns |
| 2020 | Control | 4.13 | 0.15 | 2.45 | 25.0 | 6.42 | 52.9 | 25.1 |
| | NLC | 4.02 | 0.16 | 2.47 | 25.6 | 6.58 | 54.5 | 25.7 |
| | RSM | 4.02 | 0.17 | 2.42 | 24.2 | 6.74 | 54.1 | 25.6 |
| | LCC | 3.98 | 0.17 | 2.43 | 24.5 | 6.47 | 53.9 | 26.5 |
| | $P$-value | ns | ns | ns | ns | ns | ns | ns |
| 2021 | Control | 4.00 | 0.17b | 2.41 | 24.7 | 6.60 | 48.8 | 25.1 |
| | NLC | 3.98 | 0.18b | 2.41 | 24.1 | 6.08 | 48.6 | 26.7 |
| | RSM | 4.08 | 0.20a | 2.36 | 23.4 | 6.32 | 46.0 | 25.1 |
| | LCC | 3.96 | 0.20a | 2.40 | 23.5 | 6.68 | 48.0 | 25.3 |
| | $P$-value | ns | ** | ns | ns | ns | ns | ns |

Note:
Control, no conservation practices; NLC, non-legume cover crop; RSM, rice straw mulching; LCC, legume cover crop. Different letters in each column indicate significant differences among treatments at $P < 0.01$ (**); ns, not significant.

**Table 6 Correlationship between soil physicochemical properties ($n$ = 192).**

|       | BD        | Ca       | K        | Mg    | Cu    | Fe       | Zn    | Mn  |
|-------|-----------|----------|----------|-------|-------|----------|-------|-----|
| BD    | 1         |          |          |       |       |          |       |     |
| Ca    | −0.74**   | 1        |          |       |       |          |       |     |
| K     | −0.73**   | 0.74**   | 1        |       |       |          |       |     |
| Mg    | −0.11     | 0.13     | 0.14     | 1     |       |          |       |     |
| Cu    | −0.11     | 0.02     | −0.07    | 0.10  | 1     |          |       |     |
| Fe    | −0.79**   | 0.81**   | 0.86**   | 0.19  | −0.06 | 1        |       |     |
| Zn    | −0.69**   | 0.76**   | 0.69**   | 0.11  | −0.06 | 0.82**   | 1     |     |
| Mn    | 0.22      | −0.33    | −0.26    | −0.19 | −0.01 | −0.19    | −0.17 | 1   |

Note:
** indicates a significant difference at $P < 0.01$.

practices. The concentration of $K^+$ was significantly greater by 1.1-fold in RSM and LCC treatments compared with NLC and control.

### Correlation between soil quality parameters

The BD indicated a negative significant relationship with Ca ($r = -0.74^{**}$), K ($r = -0.73^{**}$), Fe ($r = -0.79^{**}$), and Mn ($r = -0.69^{**}$). Table 6 also showed a strong positive correlation between Ca and K ($r = 0.74^{**}$), Ca and Fe ($r = 0.81^{**}$), Ca and Zn ($r = 0.76^{**}$). We found a positive very strong significant relationship between K and Fe and Mn ($r = 0.86^{**}$, $r = 0.69^{**}$, respectively). The correlation matrix also indicated a significant positive relationship between Fe and Zn ($r = 0.82^{**}$).

### DISCUSSION

Soil BD is a vital indicator of soil degradation because it influences soil porosity, plant nutrient availability, and soil microorganism activity (*Recha et al., 2022*). According to *Shaheb, Venkatesh & Shearer (2021)*, soil conservation measures decreased soil compaction, resulting in increased root development and length. *Shaheb, Venkatesh & Shearer (2021)* indicated that soil compaction reduced root biomass significantly. The decreased crop growth and yield due to soil compaction were likely due to poor nutrient availability and uptake, thus limiting/preventing root growth (*Gürsoy, 2021*). In this study, cover cropping with pinto peanut and rice straw mulch reduced BD at depths of 0–10 cm and 10–20 cm, ~0.10 g cm$^{-3}$ and ~0.08 g cm$^{-3}$ in a 3-year consecutive trial, respectively (Figs. 1A, 1B). The current research is consistent with *Mondal et al. (2019)*, who reported that conservation agriculture practices contributed significantly reduced soil compaction. Similar results have also been reported by *Degu, Melese & Tena (2019)*, *Ceylan (2020)*, and *Belayneh, Yirgu & Tsegaye (2019)*.

Like BD, soil porosity was increased significantly at two depths, 0–10 cm, and 10–20 cm, when covered with legumes and straw mulch (Fig. 2). Many studies have indicated a strong negative correlation between BD and total porosity (*Kakaire et al., 2015*; *Onwuka et al., 2020*). In the present work, cover crops and mulching decreased BD, and this may be due to reduced soil compaction, which improved total porosity. Moreover, our previous study showed that soil organic matter increased remarkably when applying cover with pinto

peanut and straw mulch (*Dung et al., 2022*). Improving soil organic carbon is the main reason for the increase in total porosity (*Fukumasu et al., 2022*).

The first year's results showed that soil nutrient concentration was not affected by soil conservation practices, except for Zn content in the depth of 10–20 cm (Table 3). However, in the second and third years, Ca, K, Fe, and Mn concentrations in RSM and LCC increased significantly at the topsoil and subsoil layers (Tables 2, 3). Conversely, these nutrients were not elevated at the depths of 20–30 cm (Table 4) and 30–40 cm (Table 5) compared with the control, except for exchangeable K at 30–40 cm in 2021. The cause may be because the root of a plant used for the cover is short, and all treatments followed the no-till practice. The results in legume biomass and straw were unable to move the depths below. Hence, it does not improve the mineral nutrients in the soil. The results did not agree with that of *Haruna & Nkongolo (2020)* that conservation practices enhanced soil nutrients in 20–40 and 40–60 cm during the second year of study. The difference in soil nutrient concentration between the two studies could be the difference in no-till and till practices. Soil conservation measures can favorably ameliorate soil fertility by enhancing the number of soil biota that decompose organic matter and, in the process, release plant-available nutrients (*Veum et al., 2015*; *Belayneh, 2019*). According to *Jat et al. (2018)*, conservation practices are considered a better alternative that recycles plant nutrients in the soil.

Our study showed soil has a high BD, which caused the availability of soil nutrients (Ca, K, Fe, and Zn) to decline. Table 6 showed that there were a strong negative correlation between BD and Ca ($r = -0.74^{**}$), BD and K ($r = -0.73^{**}$), BD and Fe ($r = -0.79^{**}$), and BD and Zn ($r = -0.69^{**}$). According to *Belayneh, Yirgu & Tsegaye (2019)*, high BD negatively affected soil nutrients due to decreased soil biological and biochemical processes, resulting in reduced soil fertility. A similar result has been reported by *Singh et al. (2020)*. They indicated a negative correlation between BD and soil nutrients. However, the results of the present work in contrast with a report of *Duan et al. (2019)*, who showed that there was a strong positive correlation of BD with exchangeable Ca ($r = 0.32$), exchangeable Mg ($r = 0.45$), and available Fe ($r = 0.71$). The results above show that the relationship between BD and soil nutrient concentration is complex.

## CONCLUSIONS

The use of soil conservation practices (LCC and RSM) significantly improved soil BD at the topsoil layer (0–10 cm) and subsoil layer (10–20 cm), enhancing soil porosity compared with applying the herbicide (control). In the first year, LCC and RSM did not affect available macronutrients (Ca, K, and Mg) and micronutrients (Cu, Fe, Zn, and Mn). However, soil nutrients (Ca, K, Fe, and Zn) increased greatly in the second and third years. The current study results suggest that farmers who cultivated fruit orchards in the VMD should use leguminous cover crops or mulch because these practices can mitigate soil compaction and degradation.

### Funding

The authors received no funding for this work.

### Competing Interests

The authors declare that they have no competing interests.

### Author Contributions

- Tran Van Dung conceived and designed the experiments, analyzed the data, prepared figures and/or tables, and approved the final draft.
- Ngo Phuong Ngoc performed the experiments, analyzed the data, prepared figures and/or tables, and approved the final draft.
- Le Van Dang performed the experiments, analyzed the data, authored or reviewed drafts of the article, and approved the final draft.
- Ngo Ngoc Hung conceived and designed the experiments, analyzed the data, authored or reviewed drafts of the article, and approved the final draft.

### Data Availability

The raw measurements are available as a Supplemental File.

### Supplemental Information

Supplemental information for this article can be found online at http://dx.doi.org/10.7717/peerj.14170#supplemental-information.

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
