# Peer review of "Impact of cover crop and mulching on soil physical properties and soil nutrients in a citrus orchard"

_PeerJ, doi:10.7717/peerj.14170_

## Round 0.1 · original submission · Major Revisions

The manuscript needs major revision. Also improve English, and try to include few updated literature of 2020, and 2021 in introduction and discussion section, thanks.

Check the pdf file attached for highlighted comments, and reply to each comment

Reviewer 1 ·

Basic reporting

The Material Method section and the Result section is written in extremely poor fashion/in a very nonscientific way. Also the english language should be rectified throughout the manuscript.I have major concerns regarding the style of writing of the result and method section. Author failed to write these two sections in a proper scientific manner.

Experimental design

Experimental design is good.

Validity of the findings

I have also concern regarding the finding of the study. Novelty is lacking in this study with the scientific content. Only the basic biochemical study has been provided. Additional molecular data should be provided to support the study.

Reviewer 2 ·

Basic reporting

This article was well written, with the aim clearly stated and supported by the review of relevant literature.
However, I have a few concerns that can be fixed to increase the value of this manuscript.
1. The manuscript has many grammatical faults. Please carefully check before resubmission. I have also carefully made several suggestions that you will find subsequently.
2. Thank you for the raw data. Firstly, the bulk density and porosity raw data shows only 0-10 cm and 10-20 cm. where is the raw data for 20-30cm and 30-40cm? For the nutrient, only raw data for 30-40cm was provided. where is 0-30cm?
It will definitely add to your manuscript quality if they were provided. Otherwise, please provide an explanation.

Experimental design

The experimental design was robust enough to investigate the aim of the study. Furthermore, the methods used in the physicochemical analyses are all acceptable, and I have no further concerns.

Validity of the findings

Apart from the few points where your results conflicted with previous studies which I have stated that you must give concrete explanations for, I have no further concerns.

Additional comments

Please see some of my suggestions

L22. change `mulched rice straw` to `rice straw mulch`.

L22. Change `cover crop by a legume` to `a leguminous cover crop`.

L42. Unclear/ambiguous sentence. Please revise.

L53. Cover cropping is NOT a crop.
`Cover cropping is a cropping system/system/method utilized to decrease erosion,......` sounds better.
or
`A cover crop is a crop utilized mainly to decrease erosion,......`

L56. `improved`.

L58. `Leguminous cover crops`.

L62. `materials`.

L68. Please delete `the`.

L68. Please delete `of soil microorganisms`.

L69. Please replace `covering crops` with `cover cropping`.

L.72. Please change `did not report` to `has not been reported`.

L84. Please delete `initial`.

L92. Please replace `are` with `were`.

L95. please replace `named` with `name`.

L97-98. The weeds were controlled when they reached about 8-10 cm tall (about 5-6 leaves).

L100-101. Please review this sentence grammatically.

L104-107. This paragraph is important. However, it needs revision.

L110. Please change `of Eijkelkamp company` to `(Eijkelkamp company)`.

L111. Please replace `to take the soil during...` with `for soil sampling in...`

Line 137. `after three years of experimentation` sounds better than in three years of experiments. Please replace it.

L138-139. This is not true because the opposite seems to be the case as the result from your Fig. 1a. showed lower bulk densities for LCC and RSM compared to the Control and NLC. Please revise and provide additional explanations.

L141-142. Please revise this sentence.
A 10-20cm BD is a bit misleading. Did you mean to say `in the 10-20cm soil depth`?

L146. Please replace with `increased greatly` with `increased significantly`.

L147. `non-legume or legume cover crops` sounds better than `cover crop by non-legume or legume`.

L149.Why not `RSM` instead of `mulched rice straw`?
There should be consistency in your reports for ease of reading and comprehension. in L139, you used RSM, so a different name here may be a bit confusing.

L160. Please replace `experiment` with `experimentation`.

L163-164. It may be better to give a more solid speculation on why Fe and Zn were `elevated greatly`.
Merely saying that the elevation is due to leguminous cover cropping and rice straw mulching is not enough.
Although the elemental content of the cover crops and rice straw mulch was not provided in this study, it is possible that their Fe and Zn contents influenced the soil concentration.

L163. Please be careful with the use of words such as `great` because it is a word that is used relatively. To avoid a conflict of understanding, keep it simple by saying `.....Fe and Zn were elevated...`

L171. Please delete `greatly`. See the previous comment for L163.

L204-205 needs revision. You probably wanted to say that - The decreased crop growth and yield as a result of soil compaction were likely due to poor nutrient availability and uptake, thus limiting/preventing root growth (Parlak & Parlak, 2011).

L205. Please replace `cover crop` with `Cover cropping`.

L206. Please replace `mulch rice straw` with `rice straw mulch`.

L214-215. `This reason may be due to reduced soil compaction,...`

L214. Please see previous comments about `greatly`.

L216-217. Please change mulched straw` to `straw mulch`.

L217. `...the main reason for the increase in...`

L219. Please change `covering crops` to cover-cropping`.

L223-224. A better explanation needs to be given for the results stated in L219-223.

L224-226. Please give your opinion or speculation why your results did not agree with the findings of Haruna and Nkongolo (2020).

L235. please delete `that`.

L2243. please change `cover crop with legume` to `LCC`.

L245. Please replace `legumes to cover crops` with `leguminous cover crops`.

Annotated reviews are not available for download in order to protect the identity of reviewers who chose to remain anonymous.

---

## Round 0.2 · accepted · Accept

The manuscript is accepted for publication

Reviewer 2 ·

Basic reporting

Thank you very much for submitting the revised version of this work. Based on my previous observations and comments, the manuscript has been comprehensively worked upon taking into consideration all of my suggestions. The grammatical errors have been rectified and the structure of the article has greatly improved. The raw data tables have also been updated and look better.
At this time, I have no further comments.

Experimental design

No further concerns on this part of the manuscript

Validity of the findings

At this point, I have no further comments regarding this area.

Additional comments

I recommend the acceptance of this work